

# Reconstruction of masked sequences *via* inverse mapping of incomplete information natural vectors

Patrick Ding[1,*], Guoqing Hu[2,3,*], Hongyu Yu[4] and Stephen S.-T. Yau[2,3,4]

[1] Whitney M Young Magnet High School, Chicago, Illinois, United States
[2] Hetao Institute of Mathematics and Interdisciplinary Sciences (HIMIS), Shenzhen, Guangdong, China
[3] Beijing Institute of Mathematical Sciences and Applications (BIMSA), Beijing, China
[4] Department of Mathematical Sciences, Tsinghua University, Beijing, China
* These authors contributed equally to this work.

## ABSTRACT

Alignment-free embedding methods, which map biological sequences into a fixed-dimensional space using mathematical techniques, hold significant value in biology. A key challenge in this field is constructing an inverse mapping to recover sequences from embedded vectors. The natural vector approach uniquely provides a theoretical one-to-one correspondence between sequences and high-order natural vectors, but reconstructing sequences from lower-order vectors remains unsolved. Moreover, when sequences contain masked regions, extracting features and constructing the inverse mapping to restore the original sequence, including the masked parts, becomes even more challenging. In this article, we define incomplete information natural vectors for masked sequences and develop a long short-term memory model that achieves over 99.9% accuracy in reconstructing unmasked positions in original sequences on SARS-CoV-2 and HIV-1 datasets, while also providing predictions for masked sites that significantly outperform random prediction. Our model can robustly handle sequences with varying masked nucleotides. Overall, our approach expands the scope of alignment-free embedding methods by enabling bidirectional conversion and addressing challenges posed by incomplete information.

Corresponding author
Stephen S.-T. Yau, yau@uic.edu

## INTRODUCTION

Advances in sequencing technologies have generated vast amounts of genomic data, ushering in the era of biological big data. As a result, sequence comparison and sequence information extraction have become pivotal tasks in biological research. Sequence comparison methods rely heavily on alignment-based techniques, which align sequences to identify similarities and differences (*Needleman & Wunsch, 1970*; *Smith & Waterman, 1981*; *Edgar, 2004*; *Higgins & Sharp, 1988*; *Altschul et al., 1990*). Sequence information extraction has seen the rise of pre-trained language models, which have proven effective in capturing complex patterns in biological sequences (*Rives et al., 2021*; *Lin et al., 2023*;

*Ji et al., 2021*). Despite advancing genomics, these methods exhibit persistent limitations. Alignment methods suffer from inefficiencies, especially when working with large datasets, and are heavily dependent on conserved regions of the sequences. Pre-trained language models, while powerful, are costly in training and lack interpretability in the resulting feature vectors.

In response to these challenges, alignment-free embedding methods have garnered significant attention as promising alternatives. These methods leverage mathematical and statistical techniques to embed sequences into a linear space, offering both efficiency and interpretability (*Yau et al., 2024*; *Zielezinski et al., 2017*; *Bonham-carter, Steele & Bastola, 2013*; *Lu et al., 2017*). They provide an attractive solution to overcome the issues associated with alignment-based methods and pre-trained language models. Within the realm of alignment-free embedding methods, a particularly intriguing problem is to build an inverse mapping that reconstructs the original sequence from its embedded vector representation. This task is crucial because it establishes a bidirectional mapping between the sequence space and the embedded linear space. The natural vector method incorporates statistical moments, converting sequences into feature information based on different k-mers and moment orders, and has demonstrated excellent performance in many fields (*Deng et al., 2011*; *Wen et al., 2014*; *Sun et al., 2021*; *Jiao et al., 2021*; *Yu & Yau, 2024b*). Natural vector embedding has a well-established mathematical proof demonstrating a one-to-one correspondence between sequences and sufficiently high-order natural vectors (*Deng et al., 2011*). However, despite its strong theoretical foundation, the reconstruction of original sequences from lower-order vectors remains unresolved. This challenge holds profound significance. Previous studies have established geometric constraints on the natural vectors corresponding to biological families using the convex hull principle (*Jiao et al., 2021*; *Zhao, Pei & Yau, 2020*). In other words, it is possible to identify natural vectors that satisfy the conditions of a given biological family. The inverse mapping can then be utilized to determine the corresponding biological sequences, enabling the prediction of undiscovered sequences in advance.

In real-world sequencing data, experimental errors often introduce positions with unknown nucleotides, referred to as masked positions. These masked regions complicate the interpretation of sequences. Many studies have focused on the issue of masking in sequences. A variety of machine learning-based methods have been employed to predict masked positions based on the surrounding context, achieving notable success (*Sun & Kardia, 2008*; *Liang et al., 2022*). More recently, pre-trained language models, inspired by BERT, have been adapted to predict artificial masks as part of their training (*Lin et al., 2023*; *Ji et al., 2021*; *Devlin et al., 2019*). However, in sequence reconstruction, the mask problem differs: our input consists solely of natural vector embeddings from masked sequences, without access to the full sequence context, and the goal is to reconstruct the entire sequence corresponding to the natural vector, including both masked and non-masked parts.

In this article, we define incomplete information natural vectors for masked sequences, thereby generalizing the use of natural vectors and demonstrating their advantages. We then embed the masked sequences and construct a long short-term memory (LSTM)

model tailored for inverse mapping. We applied our method to a masked subsequence dataset built from the genomes of the SARS-CoV-2 and HIV-1, achieving high-precision bidirectional conversion between sequences and embedding vectors. Notably, in the reconstruction from embedding vectors back to sequences, our model partially overcomes the incomplete information problem, successfully recovering many masked nucleotides. Additionally, our model demonstrates robustness: although trained solely on single-mask data, it retains partial effectiveness even in multi-mask scenarios, albeit with reduced accuracy. Our approach expands the capabilities of natural vectors and alignment-free methods, transforming unidirectional conversion into bidirectional conversion while effectively addressing the challenge posed by incomplete information due to masking.

## MATERIALS AND METHODS

### Dataset

This study utilizes the SARS-CoV-2 Dataset and HIV-1 Dataset to support our analysis. We selected RNA sequences that do not contain unknown nucleotides and have clear classification labels from public databases. (No reverse-complement sequences are included.) The SARS-CoV-2 dataset was sourced from GISAID (https://gisaid.org), including SARS-CoV-2 sequences up to June 30, 2022, while the HIV-1 dataset was obtained from the HIV Database (https://www.hiv.lanl.gov), containing HIV-1 sequences up to April 8, 2022. These sequences were split into subsequences of length 16. Each subsequence can produce 16 possible masked subsequences by replacing one known nucleotide with a masked nucleotide. We then randomly selected 2 million masked subsequences from each dataset for our analysis. For model training, each dataset was divided into three sets: 75% for training, 20% for validation, and 5% for testing. All sequence IDs corresponding to these subsequences as well as the code can be found in our GitHub repository https://github.com/PatrickDebug/LLM-reconstruct-nucleotide (DOI: 10.5281/ZENODO.16890668).

### Key terms

The definitions of key technical terms used in this article are summarized in Table 1.

### Natural vectors

The natural vector method is an alignment-free approach that converts biological sequences into vectors representing statistical moments (*Deng et al., 2011*). Consider DNA/RNA sequence $S = s_1 s_2 \ldots s_n$ and define the indicator function

$$w_k(s_i) = \begin{cases} 1, & s_i = k \\ 0, & otherwise \end{cases} \tag{1}$$

where $k, s_i \in \{A, C, G, T\}$. We define the natural vector of order $m$ of the sequence $S$ (denoted by $nv(S)$) to be

$$(n_A, n_C, n_G, n_T, \mu_A, \mu_C, \mu_G, \mu_T, D_2^A, D_2^C, D_2^G, D_2^T, \ldots, D_m^A, D_m^C, D_m^G, D_m^T) \tag{2}$$

**Table 1  Glossary of key terms.**

| Term | Definition |
| --- | --- |
| Natural vector (NV) | A sequence embedding method based on moments |
| Incomplete information natural vector (IINV) | Natural vector generalization for masked sequences |
| Recurrent neural network (RNN) | A neural network architecture for sequential data |
| Long short-term memory (LSTM) | RNN variant for long sequences |
| $n_k$ | Count of nucleotide $k$ in the sequence |
| $\mu_k$ | Mean position of nucleotide $k$ in the sequence |
| $D_j^k$ | Order $j$ information of nucleotide $k$ |

where

$$\begin{cases} n_k = \sum_{i=1}^{n} w_k(s_i) \\ \mu_k = \sum_{i=1}^{n} \frac{i}{n_k} w_k(s_i) \\ D_j^k = \sum_{i=1}^{n} \frac{(i-\mu_k)^j}{n_k^j} w_k(s_i) \\ n = n_A + n_C + n_G + n_T \end{cases} \qquad (3)$$

$n_k$ (the order 0 element) and $\mu_k$ (the order 1 element) represent the nucleotide count and the average position, respectively. $D_j^k$ (the order j element) captures higher-order information, including variance, skewness, and kurtosis. It is worth noting that the normalization constant here differs slightly from that of previous work (*Deng et al., 2011*). This modification is intended to prevent the higher-order moments from becoming too small, while still preserving the essence of the natural vector in extracting statistical moments.

The convex hull principle is an important property of natural vectors, stating that the natural vectors corresponding to biological sequences of different classifications form distinct, non-overlapping convex hulls in Euclidean space (*Zhao et al., 2019*; *Tian, Zhao & Yau, 2018*; *Yu et al., 2025*). Previous work has validated the convex hull principle in the viral genome space, demonstrating that the 32-dimensional order seven natural vector satisfies this principle (*Sun et al., 2021*). Therefore, in this study, we choose the 32-dimensional natural vector as the embedding method for sequences.

## Incomplete information natural vectors

Due to the limitations of sequencing technologies, DNA/RNA sequences obtained from sequencing often contain nucleotides that cannot be determined, which we refer to as masked nucleotides. We aim to extend the natural vector method in this scenario of incomplete information so that it can convert sequences that include masked nucleotides into natural vectors.

A straightforward method is to simply ignore the masked positions by removing all masked nucleotides and treating the remaining sequence as a regular sequence for

conversion into natural vectors. However, this trivial approach does not perform well in practice, as the information from the masked position is completely discarded.

To preserve the position information of the masked nucleotide, we introduce incomplete information natural vectors. The incomplete information natural vectors for a masked sequence are the average natural vectors of all possible complete sequences corresponding to the masked sequence. Specifically, for a sequence $S$ with $k$ masked positions, the possible complete sequences are denoted by $f_1(S), \ldots, f_{4^k}(S)$. The incomplete information natural vector of $S$ can then be represented as:

$$iinv(S) = \frac{1}{4^k} \sum_{i=1}^{4^k} nv(f_i(S)). \tag{4}$$

For example, the incomplete information natural vector of $ACNT$ can be computed as:

$$iinv(ACNT) = \frac{1}{4}(nv(ACAT) + nv(ACCT) + nv(ACGT) + nv(ACTT)).$$

The dimension of incomplete information natural vector is the same as the corresponding natural vector. That is, we consider the 32-dimensional incomplete information natural vector. In this article, we focus on subsequences with only one mask, hence $k = 1$, ensuring high computational speed.

It is worth mentioning that in real sequences, there may be many masks, which seems to imply that the summation complexity increases exponentially with the number of masks. However, this is not the case. When there are multiple masked positions, we can break the sequence into smaller pieces with each fragment containing a limited number of masks. Then we can quickly compute the incomplete information natural vector for each subsequence separately. Previous work has shown that the natural vector of the full concatenated sequence can be reconstructed through specific algebraic operations on the subsequence vectors (*Yu & Yau, 2024a*). Therefore, there is no need to compute the incomplete information natural vector for the entire sequence according to its original definition.

We can always segment the sequence in such a way that the number of masks in each subsequence is controlled, allowing us to quickly compute the incomplete information natural vector for each subsequence separately. Previous work has shown that we can directly use the operations on subsequence natural vectors to quickly compute the natural vector for the concatenated full sequence (*Yu & Yau, 2024a*), thus bypassing the issue of exponential growth in summation complexity.

In summary, incomplete information natural vectors efficiently preserve the positional information of the masked nucleotide in the overall natural vector, thereby aiding subsequent models in their analysis.

## The LSTM model

The LSTM model is employed in our study to process the data. LSTM is a specialized recurrent neural network (RNN) architecture particularly effective at modeling long-range

dependencies in sequence data, prioritizing relevant features while minimizing the impact of noise (*Hochreiter & Schmidhuber, 1997*).

In our model, the input layer accepts a 32-dimensional incomplete information natural vector derived from masked subsequences. The prediction output of the model is a matrix, which we aim to make as close as possible to the one-hot encoding of the original subsequence. The LSTM model consists of two stacked LSTM layers, each with 256 units. These layers act as feature extractors, capturing dependencies within the sequence and learning hierarchical patterns. The first LSTM layer is followed by a RepeatVector layer, which replicates the extracted features for sequence output. A TimeDistributed Dense layer with a softmax activation function is used to predict the nucleotide class (A, C, G, T) at each position in the subsequence. The model is trained using the Adam optimizer (*Kingma & Ba, 2014*) with a learning rate of 0.0005, gradient clipping, and a batch size of 32. To optimize training, early stopping and learning rate reduction callbacks were employed. Our model can efficiently run on standard desktop hardware. Benchmark tests were conducted on a system equipped with NVIDIA GeForce RTX 4090 GPU, 96 GB RAM, and AMD Ryzen 9 7950X 16-Core Processor CPU, achieving an average processing time of 17 h and 44 min for two viral analyses (2 million subsequences in each analysis).

Although using standard loss functions, such as categorical cross-entropy, is feasible, we specifically designed our loss function to emphasize the reconstruction of the masked sequence. Our loss function consists of three parts: (A) categorical cross-entropy, (B) count penalty, and (C) mask penalty. The count penalty penalizes deviations between predicted and true nucleotide counts, which is derived from a natural vector perspective. This penalty ensures that the predicted sequence is close to the true sequence by making their natural vectors similar. The mask penalty, on the other hand, applies additional penalties for errors in the masked positions, enabling the model to better reconstruct the masked nucleotides. The overall loss function is an equally weighted combination of these three components. To demonstrate that all three loss terms contribute value, we conducted experiments on a smaller dataset (200k HIV-1 viral sequences), testing each of the three loss functions (A), (B), and (C) separately, and calculated the reconstruction accuracy at masked positions. (Masked position accuracy provides better discriminative power than overall accuracy.) The results show that the combined loss achieved 34.0% accuracy, while the individual components (A), (B), and (C) achieved 33.1%, 31.5%, and 25.5% respectively, demonstrating that the combination performs better than any single component (Due to the limited dataset size, the achieved accuracy is lower than what would be expected with larger-scale training data.).

## RESULTS

### Effectiveness of incomplete information natural vectors

To demonstrate the superiority of incomplete information natural vectors over the straightforward method discussed earlier, we compared their respective L2 distances to the true natural vectors (with known masked values). In Fig. 1, using fixed-length 16 sequences, we evaluated different orders of natural vectors with 10,000 single-mask DNA sequences per order, consistently observing statistically significant advantages. Figure 2
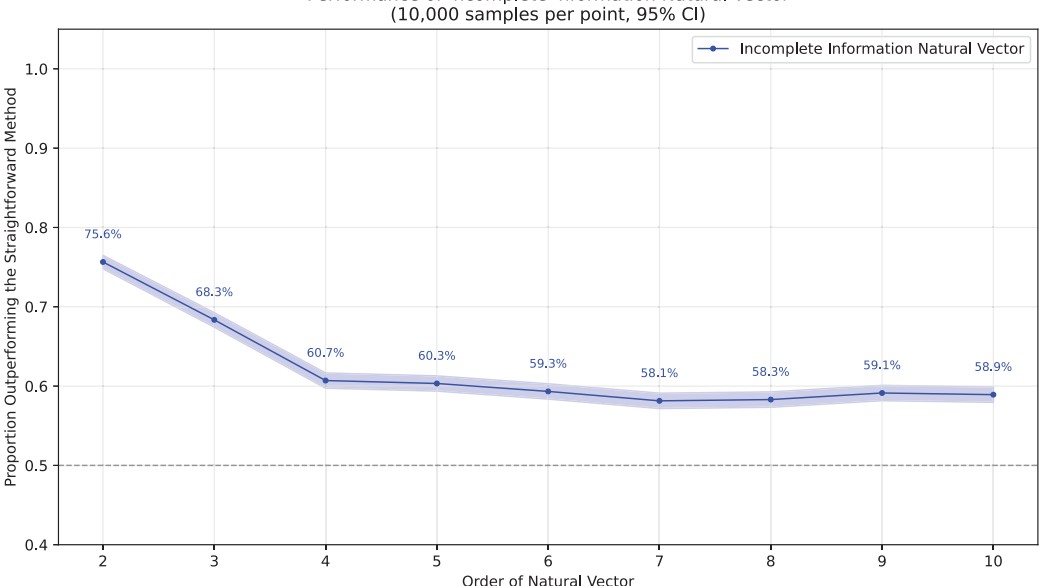

**Figure 1** Performance advantage over the straightforward method for incomplete information natural vectors across orders on single mask length-16 sequences.

shows results with fixed order seven across varying sequence lengths, testing 10,000 sequences per length containing either single masks or consecutive two-nucleotide masked regions. The incomplete information natural vectors maintained significantly smaller errors though the advantage diminished with longer sequences (converging to nearly 60% as localized masks became less influential in extended contexts). This advantage stems from the straightforward method's complete disregard of masked nucleotides' positional information, whereas our approach preserves these critical features through weighted averaging across all possible substitutions.

Furthermore, we also compared the differences in the reconstruction accuracy of the masked sequences. We conducted an analysis using HIV-1 masked sequences. Since the loss function we designed is tailored to the characteristics of incomplete information natural vectors, we used the general categorical cross-entropy as the loss function to ensure fairness. The results showed that the model using incomplete information natural vectors for feature extraction achieved an average accuracy of 95.4% across all positions. In contrast, the model using the straightforward method achieved only 81.3% accuracy. This further underscores that ignoring the positional information of the mask makes it difficult to accurately reconstruct the entire sequence.

## Masked sequence reconstruction

We transformed SARS-CoV-2 and HIV-1 masked subsequences into incomplete information natural vectors and attempted to reconstruct these sequences using the LSTM-based inverse mapping model. The results showed that 96.3% of nucleotides for SARS-CoV-2 and 96.2% for HIV-1 were correctly reconstructed by the model, which is a

Peerj

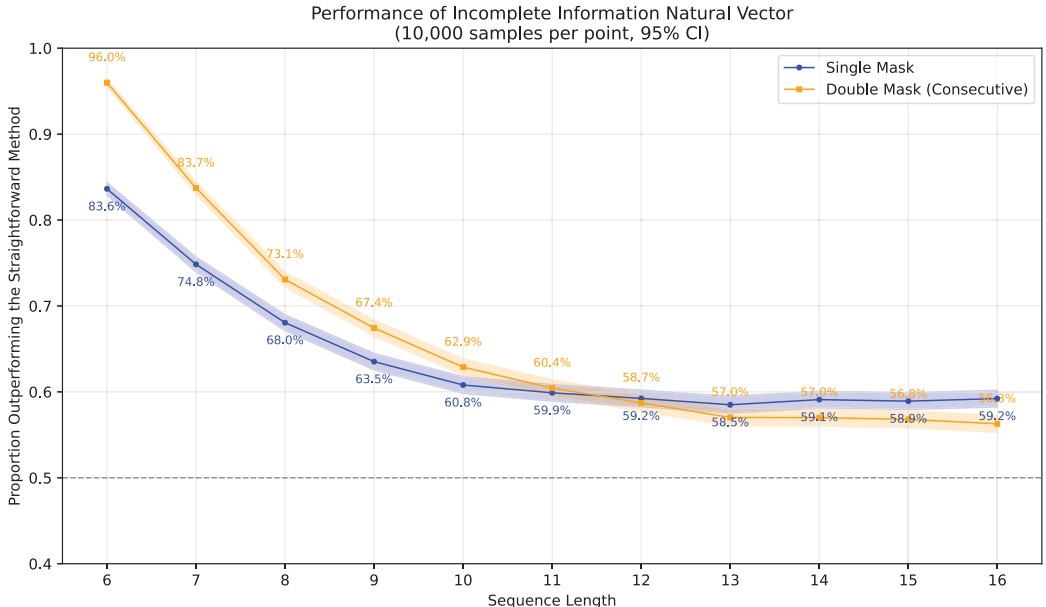

**Figure 2** Performance advantage over the straightforward method for incomplete information natural vectors across sequence lengths.

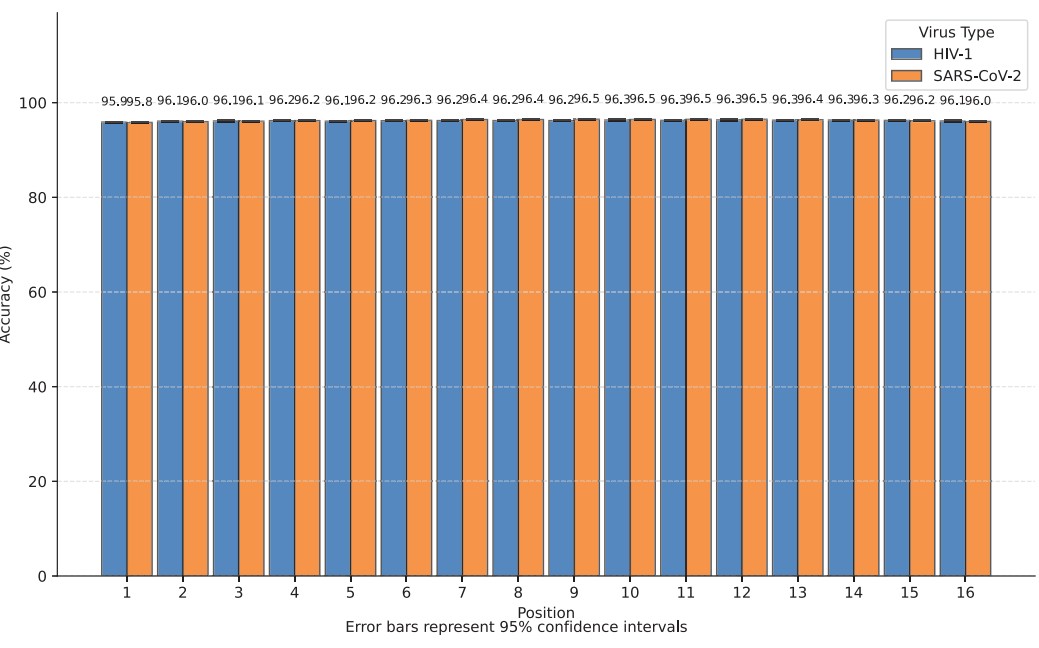

**Figure 3** Bar graph of reconstruction accuracy for all positions.

very high accuracy rate. In Fig. 3, we displayed the accuracy for the 16 positions of the subsequence.

We observed that nearly all inaccurate reconstructions occurred at the masked positions. For non-masked positions, the accuracy of the inverse mapping exceeded 99.9%

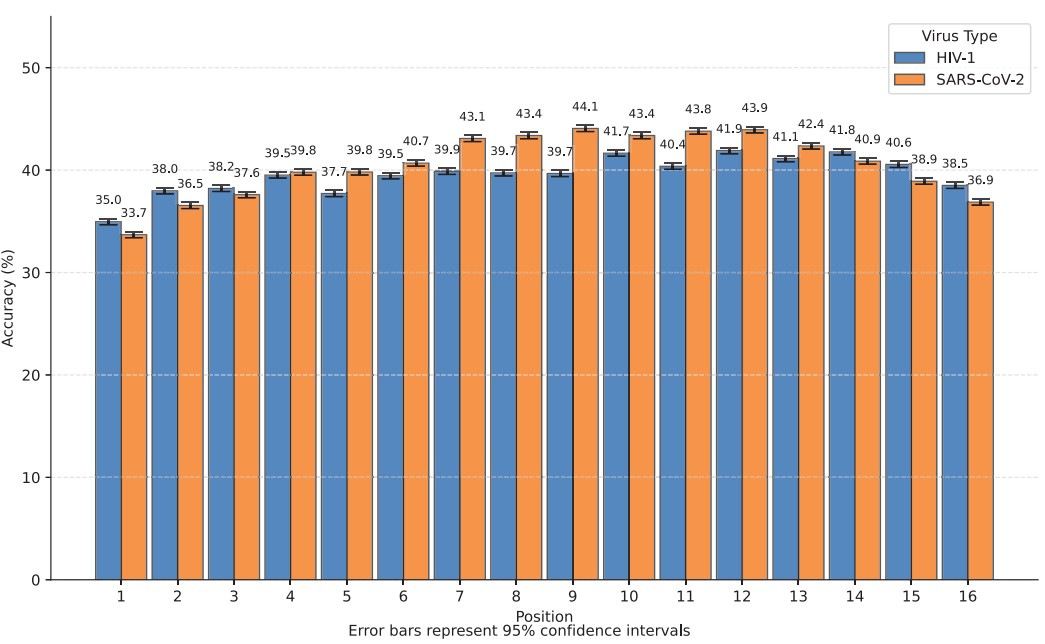

**Figure 4  Bar graph of reconstruction accuracy for masked positions.**

(99.98% for SARS-CoV-2 and 99.97% for HIV-1). For the nucleotides at the masked positions, the reconstruction accuracy was 40.6% for SARS-CoV-2 and 39.6% for HIV-1. In Fig. 4, we showed the reconstruction accuracy for the masked nucleotides at the 16 positions. We saw a statistically significant trend where the accuracy for nucleotides in the middle is higher compared to those at the ends. This phenomenon likely occurs because central nucleotides receive bidirectional sequence context information, whereas terminal nucleotides only have unidirectional contextual information available (This pattern appears less distinct in Fig. 3 due to the inclusion of unmasked positions where accuracy approaches 100%, creating a ceiling effect that compresses the visible variation.). If we retrained the model on sequences with masks fixed at the two central positions (positions 8 or 9), the mask prediction accuracy will improve to 47.0% for SARS-CoV-2 and 52.2% for HIV-1 (In this task, to augment the limited training data, we utilized all available subsequences from the SuppID.xlsx file on GitHub, lifting the original 2-million subsequence restriction.).

The fundamental performance limitation stems from the task's inherent ambiguity: the dataset contains distinct authentic sequences whose masked subsequences are identical. For instance, consider a sequence fragment $s_1$ that undergoes a single-nucleotide mutation to become $s_2$. If the mask occurs precisely at the mutation site, we cannot determine whether the original sequence was $s_1$ or $s_2$. While both reconstructions are theoretically valid, only one would be counted as correct when calculating mask reconstruction accuracy, thereby imposing an upper bound on the achievable accuracy.

## Robustness of the inverse mapping

The proposed inverse mapping model demonstrated strong robustness against input variations. Although trained exclusively on sequences containing a single masked element, the model successfully reconstructed original sequences when processing natural vectors derived from both unmasked sequences and sequences with two masked elements without requiring parameter adjustments. It is worth noting that while retraining the model would likely achieve higher accuracy, we have focused on demonstrating the results obtained without modifying model parameters, solely through adjustments to the input format. Specifically, for standard natural vectors from unmasked sequences, reconstruction accuracies reached 99.98% and 99.95% on SARS-CoV-2 and HIV-1 datasets, respectively. Additionally, when handling incomplete information natural vectors from double-masked sequences, the model maintained accurate reconstruction with accuracies of 84.9% and 85.0%, respectively (These results were derived from 10,000 sequences in the test set with re-masked elements.). These results highlighted two critical advantages. First, the model generalizes effectively to process standard natural vectors despite being trained on incomplete information natural vectors. Second, the model exhibits enhanced tolerance to twice the noise complexity encountered during training. These results underscore the model's adaptability to diverse real-world scenarios where input completeness and noise levels may vary unpredictably.

## DISCUSSION

The natural vector method, due to its theoretical and practical advantages, has been widely applied in a range of fields such as sequence feature extraction, sequence generation, and phylogenetic analysis (*Deng et al., 2011*; *Jiao et al., 2021*; *Yu & Yau, 2024b*). However, most natural vector processing methods focus on high-quality, mask-free sequences to extract genuine information. In contrast, this work defines incomplete information natural vectors to extract as much information as possible from masked sequences, and subsequent inverse mapping is used to verify that this extraction does not lead to excessive distortion despite the incomplete information. By utilizing incomplete information natural vectors, we are able to validate a series of experimentally verified properties of natural vectors, such as the convex hull principle, on a broader masked dataset, thereby generalizing the value of these properties.

In materials and methods, we discussed the relationship between the incomplete information natural vectors of subsequences and those of the complete sequences. The concatenation of sequences can be expressed through simple operations on natural vectors, which means that by efficiently computing the incomplete information natural vectors for subsequences with few masks, we can quickly obtain the incomplete information natural vectors for the full sequence, even if it contains many masks (*Yu & Yau, 2024a*). This strategy circumvents the computational complexity that would otherwise grow exponentially with the number of masked positions in full-length sequences. Therefore, by focusing on subsequences, we are able to cover the general case without losing generality.

We successfully constructed an inverse mapping for masked sequences using incomplete information natural vectors, achieving an accuracy exceeding 99.9% in unmasked positions and maintaining over 96% overall reconstruction accuracy across all positions. Our model exhibits strong robustness. Although it was trained solely on single-mask data, it reliably processes natural vectors from sequences with no masks as well as those with double masks. These results underscore the model's adaptability in real-world scenarios.

It should be emphasized that our model differs fundamentally in its objectives from conventional mask prediction approaches. Traditional mask prediction methods, typically based on BERT-style architectures, utilize contextual sequences as input to predict the original values at masked positions. In contrast, our approach employs embedded feature vectors as input to reconstruct the original pre-embedded sequences. During this reconstruction process, our method can overcome the limitations of incomplete information to partially recover masked values—thus, mask prediction emerges as a byproduct rather than the primary objective of our framework. The true significance of our work lies in the inverse mapping itself as well as its ability to overcome the limitations introduced by incomplete information.

From a theoretical standpoint, the inverse mapping enables the reconstruction of the original sequence from its embedded vector. This capability confirms that critical sequence information is not lost during the embedding process, thereby validating the effectiveness of our chosen feature extraction method. Furthermore, by bridging the gap between the vector space and the sequence space, we convert the traditionally one-way mapping from sequences to features into a bidirectional process.

In practical terms, the most important application stems from the convex hull principle. The convex hull principle states that the natural vectors corresponding to sequences from different biological families form mutually disjoint convex hulls in space, providing geometric constraints on sequences within a specific family in the natural vector space. This allows us to identify new natural vectors that satisfy the conditions of a given biological family (*Jiao et al., 2021*). With the construction of the inverse mapping, we can further discover potential new sequences. Compared to generative models, this approach offers fundamentally improved interpretability, making it highly promising for future research.

## CONCLUSION

In summary, this article defines the incomplete information natural vector for masked sequences, a novel approach that enables effective feature extraction despite missing data, and leverages it to construct a highly accurate and robust inverse mapping. While this mapping helps overcome the challenges of incomplete information and achieves competitive masked sequence prediction, its primary significance lies in bidirectionally bridging sequence space and feature space. This achievement rigorously validates the preservation of sequence information integrity during the feature embedding process. By integrating with the convex hull principle, this approach can provide a novel interpretable theoretical framework for discovering biologically functional sequences.

### Funding
This work is supported by the National Natural Science Foundation of China (No. 12171275), and Tsinghua University Education Foundation. The funders had no role in study design, data collection and analysis, decision to publish, or preparation of the manuscript.

### Grant Disclosures
The following grant information was disclosed by the authors:
National Natural Science Foundation of China: 12171275.
Tsinghua University Education Foundation.

### Competing Interests
The authors declare that they have no competing interests.

### Author Contributions

- Patrick Ding conceived and designed the experiments, performed the experiments, analyzed the data, prepared figures and/or tables, authored or reviewed drafts of the article, and approved the final draft.
- Guoqing Hu conceived and designed the experiments, analyzed the data, authored or reviewed drafts of the article, and approved the final draft.
- Hongyu Yu conceived and designed the experiments, analyzed the data, prepared figures and/or tables, authored or reviewed drafts of the article, and approved the final draft.
- Stephen S.-T. Yau conceived and designed the experiments, authored or reviewed drafts of the article, and approved the final draft.

### Data Availability
The data is available both at GitHub and Zenodo:
- https://github.com/PatrickDebug/LLM-reconstruct-nucleotide.
- PatrickDebug. (2025). PatrickDebug/LLM-reconstruct-nucleotide: Initial Release (InitialRelease). Zenodo. https://doi.org/10.5281/zenodo.16890669.

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
