# Peer review of "Reconstruction of masked sequences via inverse mapping of incomplete information natural vectors"

_PeerJ, doi:10.7717/peerj.20126_

## Round 0.1 · original submission · Major Revisions

Two experts in the field reviewed your manuscript. As you can see from their comments below, both raise several points for further improvements of the manuscript. Please read their comments carefully and revise the manuscript accordingly. I think it is essential to provide more details about their method, including the release of the code.

**Language Note:** The review process has identified that the English language must be improved. PeerJ can provide language editing services - please contact us at [email protected] for pricing (be sure to provide your manuscript number and title). Alternatively, you should make your own arrangements to improve the language quality and provide details in your response letter. – PeerJ Staff

Reviewer 1 ·

Basic reporting

1. While the overall language is generally clear, there are several instances of awkward grammar or phrasing. For example, in line 36, the phrase “are lack of interpretability” should be corrected to “lack interpretability”.
2. Figures 2 and 3 lack sufficient annotations and legends. It is unclear whether the differences across positions are statistically significant. Including confidence intervals or standard deviations would improve interpretability.

Experimental design

3. The loss function includes custom terms like the count penalty and mask penalty, but the manuscript does not provide quantitative evidence that these improve model performance. Would it be possible to include an ablation study to evaluate their individual contributions?
4. Although the model was trained only on single-masked subsequences, it is also used to reconstruct double-masked sequences. Has the generalizability of this approach been sufficiently validated? Is there any overlap between training and test data that might artificially inflate performance?
5. Are the masked positions uniformly distributed in the training data? Could the observed position-specific differences in reconstruction accuracy be due to imbalance in training frequency?

Validity of the findings

6. The high performance on the test set could be influenced by the very large training set. Could the authors include results on an independent or cross-species test set to better assess generalizability?
7. The manuscript distinguishes this method from BERT-style models, but the accuracy for masked nucleotide prediction is still relatively modest compared to state-of-the-art transformers. What types of biological applications is this method most suited for?
8. The work convincingly shows the utility of natural vectors under incomplete information. However, could the authors evaluate its robustness under more realistic sequencing error patterns, such as indels or longer ambiguous regions?

Additional comments

9. The extension of the natural vector framework from feature extraction to bidirectional sequence reconstruction is conceptually sound and adds theoretical depth. Nevertheless, the absence of code or scripts is a significant limitation. For a methods-focused manuscript, sharing reproducible code is essential. I recommend providing a GitHub link or at least core code snippets in the supplementary material. Moreover, the figures could benefit from stronger visual storytelling, such as highlighting confidence levels for masked position predictions or visualizing prediction entropy across positions.

·

Basic reporting

There is some proofreading needed with minor grammatical and or syntax errors. Some examples:
Line 37: “are lack of interpretability” → “lack interpretability”.
Line 64: “without the complete contextual sequence” could be more clearly stated as “without access to the full sequence context”.

Furthermore, the paper would benefit from a short glossary or table to define key terms like “natural vector” and “incomplete information natural vector.” This would be useful for readers not familiar with this methodology. This would also help explanations for terms such as Dkj, which are introduced without definitions, Line 92-96.

Experimental design

Line 70-74: Some discussion about the methods' application to real sequencing datasets with Ns or ambiguous calls is essential. Include limitations: e.g., performance drop when noise increases or sequence lengths vary widely.
Line 84: This section should include information about whether or not reverse-complement sequences can be used
Lines 122-123: This is potentially a useful approach. However, it should be explicitly stated that when there are multiple masked positions, the model used will break the sequence into smaller pieces with each fragment containing a single masked site.
A worked example would be useful to demonstrate how the IINV is calculated. This would ideally be put into the section Lines 105-115.
Line 138-9: Details of time taken and computational tools (RAM, GPU, etc) would be very helpful at this point.
Line 148-9: The weighting is vague. Please provide more details of how these weights are assigned.

Validity of the findings

This paper presents a methodologically sound approach to reconstructing masked biological sequences using incomplete information natural vectors (IINVs). Several limitations should be noted. First, although the reconstruction accuracy for unmasked positions exceeds 99.9%, the accuracy at masked sites remains relatively low (~40–52%), which—while better than random—may limit utility in applications where these positions are biologically critical. Second, the method’s reliance on global vector representations means it does not leverage local sequence context as effectively as pre-trained language models, potentially reducing reconstruction quality in complex genomic regions. Third, the model is validated on short 16-mer sequences; while the authors suggest theoretical scalability via subsequence concatenation, practical performance on longer sequences is not demonstrated. Finally, while the theoretical use of the convex hull principle is compelling, its practical application (e.g., discovering novel sequences) is not fully explored. These issues do not detract from the core contributions of the study but suggest important directions for further development and validation.

Particular updates required are:
The authors acknowledge that some distinct sequences can yield identical masked subsequences (line 185). This ambiguity needs further discussion—how might this impact downstream applications such as diagnostics or phylogenetics?

Line 226-228: Expand on this difference—it is important for the reader that the authors highlight that models like BERT take full context, while this model does not, and why this distinction matters in practice.

---

## Round 0.2 · accepted · Accept

Since neither of the previous reviewers responded to the invitation to re-review, I myself confirmed that you have addressed all of their points. Thus, I am pleased to accept the current manuscript. Congratulations!